# Exploring data literacy self-perception among Indonesian high school students

**Charanjit Kaur**[1]*, **Pei P. Tan**[1], **Nurjannah Nurjannah**[2], **Ririn Yuniasih**[1]

1 Monash University, Melbourne, Australia, 2 Brawijaya University, Malang, Indonesia

* charanjit.kaur@monash.edu

**Data Availability Statement:** All relevant data are within the manuscript and its Supporting Information files.

**Funding:** The author(s) received no specific funding for this work.

## Abstract

Data is becoming increasingly ubiquitous today, and data literacy has emerged an essential skill in the workplace. Therefore, it is necessary to equip high school students with data literacy skills in order to prepare them for further learning and future employment. In Indonesia, there is a growing shift towards integrating data literacy in the high school curriculum. As part of a pilot intervention project, academics from two leading Universities organised data literacy boot camps for high school students across various cities in Indonesia. The boot camps aimed at increasing participants' awareness of the power of analytical and exploration skills, which in turn, would contribute to creating independent and data-literate students. This paper explores student participants' self-perception of their data literacy as a result of the skills acquired from the boot camps. Qualitative and quantitative data were collected through student surveys and a focus group discussion, and were used to analyse student perception post-intervention. The findings indicate that students became more aware of the usefulness of data literacy and its application in future studies and work after participating in the boot camp. Of the materials delivered at the boot camps, students found the greatest benefit in learning basic statistical concepts and applying them through the use of Microsoft Excel as a tool for basic data analysis. These findings provide valuable policy recommendations that educators and policymakers can use as guidelines for effective data literacy teaching in high schools.

## Introduction

Data is becoming increasingly ubiquitous today, and data literacy is an essential skill in learning and employment. It is a fundamental skill because more organisations rely on data-driven decision-making. The Organisation for Economic Co-operation and Development [1] emphasises the importance of data literacy for learning in 2030 and beyond as it equips students with "the fundamental conditions and core skills, knowledge, attitudes and values that are prerequisites for further learning across the entire curriculum" (p.2). Data literacy is the ability to read, comprehend, analyse and communicate data [1–3]. Students must be able to think critically while working with data and interpret the generated statistics as evidence. For students in secondary education, in particular, data literacy is crucial to pursue further education and future career [2, 4–6]. With the proliferation of data in the modern world, students must learn how

**Competing interests:** The authors have declared that no competing interests exist.

to assess the validity and reliability of sources, as well as how to use data to draw meaningful conclusions.

Despite increasing awareness of the importance of understanding data in a broader scope, both in generating and using data, the initiatives to address this issue still need to be improved [5]. Current research shows that "literature and skill training on data literacy has been slow to develop" [7], and there is still much space for new initiatives in this discipline area [7, 8]. Most data literacy initiatives in secondary education have been focused on building students' numeracy skills to improve their capacity to work with quantitative data [5]. However, the initiatives have not adequately addressed the primary emphasis of data literacy teaching in the Secondary curriculum to ensure students grasp basic statistical concepts and their real-world applications. The results of a study on data literacy in secondary science classrooms show that although students are competent in data exploration skills, they struggle with applying these skills. This is especially true when evaluating the data and creating evidence to support their arguments [9]. Similar findings were reported from a study of data literacy in Physics among Indonesian high school students [10]. According to the study, students have difficulties assessing data quality and analysing, interpreting, implementing and evaluating data. As a result, they are not able to utilise subject-specific data to address real-world problems [10].

The gaps identified from existing research and literature on data literacy indicate a greater need for innovative initiatives when teaching data literacy skills in high schools. Research suggests that integrating data literacy skills in specific subjects, such as physics, along with relevant learning materials is necessary to improve students' skills in collecting, processing and analysing data [10]. Furthermore, most of the research on data literacy research is conducted from the perspective of researchers rather than that of participants [8]. Therefore, it is crucial that we explore the view of students when generating new insights into data literacy studies.

Within the context of education in Indonesia, the government's new Education initiative prioritises the inclusion of data literacy in the high school curriculum. According to the Indonesian Minister of Education, Culture, Research and Technology, Nadiem Makarim, to accelerate educational innovation in Indonesia, young Indonesians must acquire data and computer literacy, including statistics, coding, and programming [11]. The new roadmap introduced by the Minister includes an Emancipated Learning program known as *Merdeka Belajar* [12]. *Merdeka Belajar* outlines the need for a curriculum based on competencies. It provides teachers and students with the autonomy to improve their teaching and learning, with a greater emphasis on technology and innovation [12].

To support the government's efforts, there needs to be greater awareness of the criticality of data literacy skills. We designed a pilot intervention project that complements *Merdeka Belajar*. The intervention comprises data literacy boot camps collaboratively organised by academics from Monash University and Universitas Brawijaya. Students in year 10 and year 11 from high schools (*SMA*: *Sekolah Menengah Atas*) and vocational high schools (*SMK*: *Sekolah Menengah Kejuruan*) from different regions in Indonesia were invited to participate in the boot camps, aiming to accommodate students from different disciplines. The participants attended two virtual boot camps, two-hours long in each session, in which they learned some basic statistical concepts and its applications in daily life. Case studies using simple data sets, such as analysing hospital quality and employees' performance, were provided to help students to apply their understanding of basic statistical concepts such as mean, median, and standard deviation.

The boot camps were developed following the framework outlined in the Pre-K–12 Guidelines for Assessment and Instruction in Statistics Education II (GAISE II), published by the National Council of Teachers of Mathematics/NCTM [13]. These guidelines emphasise the importance of teaching statistics as an inquiry-based process. In designing the boot camps, we

recognised the need to focus on statistical thinking and conceptual understanding when teaching introductory statistics courses in colleges [14]. Equipping students with fundamental statistical concepts and their applications is crucial for a smooth transition from secondary to higher education.

This paper investigates how high school students perceive data literacy after participating in the boot camps. Qualitative and quantitative data were collected through student surveys and focus group discussions with students to capture how students perceive their ability of basic statistical concepts, data analysis, data presentation, data rationalisation and basic Excel skills. We also explored what is currently lacking at schools, students' learning aspirations and the learning materials they find valuable and helpful for their development. The findings provide policy recommendations for educators and policymakers when implementing data literacy teaching strategies.

## The framework for the study

The framework for the pilot intervention project is twofold, considering project participants are high school students transitioning into higher education. Firstly, we looked at current levels of data literacy and skills taught at participating schools. We adopted strategies proposed by Pre-K–12 Guidelines for Assessment and Instruction in Statistics Education II (GAISE II) by the National Council of Teachers of Mathematics [13] in order to enhance their skills.

In the second fold, we designed learning materials for boot camps that focused on cultivating data literacy skills, statistical thinking and conceptual understanding to prepare students for tertiary education. The teaching strategies for the boot camps were drawn from the *Guidelines for Assessment and Instruction in Statistics Education (GAISE) College report 2016* [14].

Based on the two guidelines above, we designed the learning materials based on a problem-solving approach. This approach requires students to acquire the ability to understand and solve investigative statistical questions, collect and analyse data as well as interpret results. These components are depicted in Fig 1 below.

We also included a data challenge for participants to engage in "an investigative process of problem-solving and decision making" [14] (p.3) and "experience with multivariable thinking" [14] (p.3). In the data challenge, we used real-world data with a particular context and purpose [14] so that students would better understand basic statistical concepts learned at school.

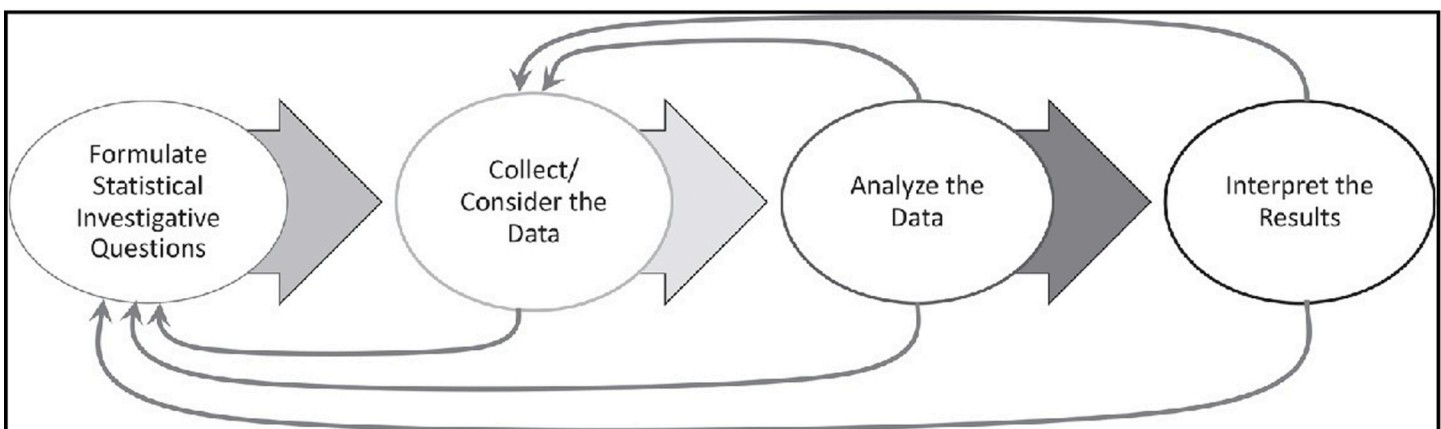

**Fig 1. Statistical problem-solving process, adopted from the Pre-K–12 Guidelines for Assessment and Instruction in Statistics Education II (GAISE II), by the National Council of Teachers of Mathematics, 2020.**

To support student learning before and during the boot camps, we designed a Google site as a centralised repository for learning materials used during the boot camps. The site contained short videos in preparation for the boot camp activities. Research shows that videos can be an effective learning tool [15] as they are easily accessible, provide visual aid and help keep students engaged. In addition to videos, the Google site housed all data sets and activity sheets used during the boot camps. The framework described above sets the parameters we used to develop the instruments for data collection, including surveys and questions for focus group discussions.

## Methodology

A mixed method research design is used in this pilot intervention by combining or integrating both quantitative and qualitative data in a single study. It was conducted using a convergent parallel mixed-methods approach in which quantitative and qualitative data were collected within the same survey [16, 17]. This approach aimed to capitalise strengths of both types of data. The qualitative and quantitative data were analysed separately, and then combined to draw general findings of the study for further discussion [16, 17], that allowed this study to develop richer insights by combining information obtained from both quantitative and qualitative data.

As the primary data collection method, online surveys were distributed among students before and after the boot camps. In these surveys, quantitative data was generated from students' responses to a series of questions asking their self-perceptions towards their data literacy skills before and after the boot camps. The same surveys generated qualitative data from students' answers to open-ended questions about what students know and want to know about data literacy. Details of the survey are provided in the data collection section. Additional qualitative data was generated from a focus group discussion (FGD) that aimed to elaborate on students' responses in the surveys and explore their experience and opinions toward the data literacy boot camps. In the final part of the surveys, students were asked to indicate their agreement to join the FGD. Participants of the FGD were students who were confirmed and available to join the FGD at a mutually agreeable time. Questions during the FGD focused on exploring how the boot camps raised participants' awareness of the importance of data literacy skills, as well as gathering their opinions and feedback on the boot camps.

## Participants

Participants of this pilot project were students of year 10 and year 11 from SMA and SMK from different regions in Indonesia who attended the data literacy boot camps. Initial invitation to participate in the project was circulated through a network of teachers, which one of the research team members has an access to. From this initial invitation, teachers from five different schools indicated their agreement to participate in the project, then formal invitations were sent to the schools respectively. Permission to participate in the research project was sought from the school principals. Apart from acknowledging students' participation in the project, the principals also gave permission to students to use the schools' computer labs to participate in the data literacy boot camps, with supervision of the teachers. The use of the school computer lab was mainly because not all students owned personal computers or laptops, which were needed for the boot camps that were delivered online. Also, with supervision of the teachers, students would have support in case they experienced some technical difficulties during the sessions in the boot camps.

Teachers from participating schools shared information about the project to their students in year 10 and year 11 in person and called for voluntary participation. Students who expressed

interest were given written explanatory statements and informed consent was sought from students and their parents prior to the boot camps. Ninety-seven students participated in this pilot project, and they were located in different cities and islands of Indonesia. Considering the demography of participants, the boot camps were conducted virtually, allowing participation from different locations [18]. Details of participants is described in the Table 1 below.

The Table 1 above shows that 58 of the participants were from SMA and 39 from SMK, and approximately 60% of students from SMAs are male while more than two-thirds of those from SMKs are female. Distribution of participants by gender and school is shown in the Fig 2.

The student participants were from a range of disciplines, depending on whether they were from SMA or SMK schools. This is depicted in the Fig 3 below.

The majority of SMA students were from maths and science, while a large proportion of those from SMK were from Banking and Microfinance, followed by Accounting. The diverse background of participants' areas of study, allowed us to explore the relevance of data literacy skills to specific disciplines in secondary education.

## Data literacy boot camps

As a central element of this pilot intervention project, data literacy boot camps were conducted with the primary purpose to increase awareness among participants about the importance of analytical and exploration skills. Through two sessions of virtual boot camp, students were provided with hands-on and engaging learning activities covering basic concepts of statistics, types of data, data analysis, data rationalisation, and data visualisation. Since the boot camp was using Microsoft Excel as a tool for basic data analysis, it also provided opportunities for students to improve their basic excel skills.

A team of academics from Monash University and Universitas Brawijaya collaboratively work together in developing learning materials for the boot camps. With reference to the statistical problem-solving process that was adopted from the GAISE report [14] as the framework of the study, the materials focused on developing students' skills on statistical thinking and conceptual understanding that would help them prepare for tertiary education. The learning materials included instruction videos, activities sheets with step-by-step instructions, data sets and data challenges, all were organised in a Google site. The use of video instructions allowed audio visual engagement to the content that facilitated cognitive process and language comprehension [15]. This was particularly beneficial for the boot camp, having video instructions and other learning materials online made them accessible for the participants who were

**Table 1. Demographic data of the boot camp participants.**

| Demographics | SMA (Sekolah Menengah Atas) High Schools | SMK (Sekolah Menengah Kejuruan) Vocational High Schools |
|---|---|---|
| Year 10 | 27 | 19 |
| Year 11 | 31 | 20 |
| Male | 34 | 9 |
| Female | 24 | 30 |
| Urban * | 21 | 14 |
| Sub-urban* | 37 | 25 |
| Total number of students | 58 | 39 |

*Based on geographical location of participating schools

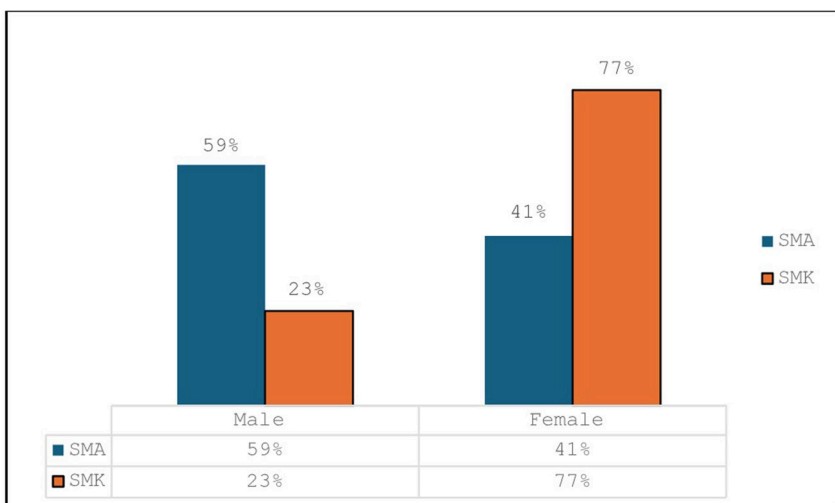

**Fig 2. Distribution of students by gender and school.**

located in different areas. Access to the site was given prior to the boot camp so participants could get familiar with the contents and had better preparation for the boot camp.

Teachers from participating schools were involved in planning the boot camps, considering the critical role of teachers in developing students' data literacy skills [19]. An initial meeting was conducted to discuss the details of the boot camps with the teachers and to gain an understanding of what has currently been taught at school. Information from teachers was instrumental to understand the starting points to scaffold participants' data literacy skills through the boot camp. Involving teachers in the development of data literacy boot camp programs aimed to ensure that the contents were relevant, aligned with educational agenda, and met the needs of both students and the school [19]. This approach was also aligned with the first fold of the framework of the study to look at current levels of data literacy and skills taught at participating schools [13] in order to enhance students' skills.

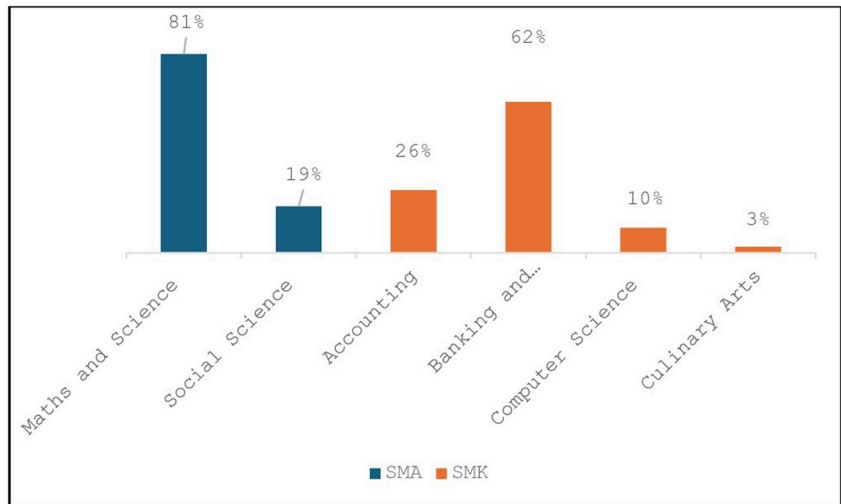

**Fig 3. Distribution of disciplines by school.**

Another important input from teachers was related to the delivery of the boot camps, as it was expected to not interfere with the school programs and activities. According to the input, the boot camp was conducted after the-end-of-semester exams, so it would not conflict with exam preparation. The boot camp was delivered in two sessions, each session lasting for two hours. The sessions were facilitated by the same academic team who developed the learning materials.

## Data collection

A Likert scale-based questionnaire was designed to measure students' perceptions towards their data literacy skills before and after the boot camps. Since data literacy skills were crucial for navigating the data-driven landscape, key indicators or instruments to assess these skills involved various aspects of students' perception and comprehension of data, practical skills and attitude towards data [20]. Also, referring to the second fold of the framework of this study [14], the questionnaires aimed to capture students' perceptions of data literacy skills, statistical thinking and conceptual understanding.

The questionnaire was divided into the following categories: Basic statistical concepts, Excel skills, Data analysis skills, Perception of data literacy, Data rationalisation skills. The basic statistical concepts involved the understanding of data types, descriptive statistics, sampling method, data cleaning and data tidying. The assessment of basic excel skills involves the understanding of using basic formula, pivot table and vlookup function. The instruments cover how the attitude of the students towards data such as understanding what data literacy entails and its importance is crucial. This includes recognizing the ability to explore, understand, and communicate with data. The last instrument related to data rationalization skills which involves organizing, cleaning, and structuring data for effective use to ensure data quality and consistency.

Pre-surveys were distributed on 24 June 2022 to gather students' perception on their data literacy skills prior to the boot camp. After the boot camp, post-surveys were distributed on 17 July 2022 to generate students' perception after joining the boot camp. When measuring changes in student perceptions towards their data literacy skills, we filtered responses to include students who completed the questionnaires before and after the boot camp. Approximately forty per cent of all participants were included in our data.

In addition to the student survey, we conducted a focus group discussion with three male and two female students. This discussion aimed to explore participants' experiences and opinions about the boot camp. It allowed them to elaborate on their responses to the survey. This method is useful in generating information that can "construct descriptive and interpretive accounts of participants' experiences and meanings" [21] (p.1210). Semi-structured guiding questions were designed for the group discussion in order to explore participants' experiences and opinions about the boot camp. Five students, three males and two females, participated in the hour-long Zoom discussion. Transcripts of the discussion were then analysed to explore themes that emerge from participants' responses to the questions. The finding from this thematic analysis was then interpreted within the context of high school education in Indonesia. Practice and meaning should be understood within "the contexts in which they occur" [21] (p.1210). Furthermore, in order to understand parts of a phenomenon, we need to have a grasp of the entire phenomenon.

## Analysis and results

The following statistical analysis was carried out using the R package. The Cronbach's alpha reliability test assessed the questionnaires' reliability and consistency. In addition to

Cronbach's alpha, omega coefficients were used to test the consistency of the questionnaires, in which Cronbach's alpha assumption fails with unequal tau-equivalence or error variance. Because the questionnaires were on a 5-point Likert scale, the difference between the pre-post survey measures for each variable was tested using the Wilcoxon signed rank test. The magnitude of difference for the pre-post study was also examined using effect size. Furthermore, the size of the effect acts as an indicator of the workshop's effectiveness.

The statistical analysis was performed using the R software package. To evaluate the reliability and internal consistency of the questionnaires, Cronbach's alpha was calculated. This test is commonly used to determine how well a set of items measures a single unidimensional latent construct. However, Cronbach's alpha assumes that all items contribute equally to the total score and equal error variance, which may not always hold true. To address potential violations of these assumptions, omega coefficients were also computed. Omega is a more flexible measure that provides a better estimate of reliability in cases where the assumptions of Cronbach's alpha are not met.

Given that the questionnaires employed a 5-point Likert scale, which produces ordinal data, the Wilcoxon signed-rank test was used to compare pre- and post-survey measures. This non-parametric test is appropriate for detecting differences in paired data when the assumptions of parametric tests, such as normality, are not satisfied. To quantify the magnitude of change between the pre- and post-intervention surveys, effect size was calculated. The effect size helps to assess the effectiveness of the workshop by indicating how meaningful the observed changes were.

## Results

Forty-one students participated in and completed pre-post Likert scale questionnaires about their perceptions of data literacy from various perspectives. Cronbach's alpha values for the pre-post survey on each data literacy component are shown in Table 2. The Cronbach's alpha values are in the acceptable range, with a minimum of 0.96 for the post-survey and a minimum of 0.94 for the pre-survey. There is sufficient internal consistency, as indicated by the alpha and omega values, suggesting no one-dimensional questions exist.

Fig 4 depicts the density distribution of the total score for the pre- and post-survey based on students' perceptions of different categories (basic statistical concept, Excel skill, data analysis, perception of data literacy, and data rationalisation). The density distribution shows that the student's perceptions about their understanding of basic statistical concepts and data analysis

**Table 2. Descriptive statistics and reliability tests.**

| Survey | Components | Mean | S.D. | $\alpha$ | $\omega$ |
|---|---|---|---|---|---|
| Pre | Basic Statistical Concept | 2.77 | 1.23 | 0.97 | 0.98 |
| Post | | 3.33 | 1.15 | 0.97 | 0.98 |
| Pre | Basic Excel Skill | 3.43 | 1.34 | 0.94 | 0.97 |
| Post | | 3.76 | 1.11 | 0.97 | 0.98 |
| Pre | Data Analysis | 2.70 | 1.28 | 0.94 | 0.97 |
| Post | | 3.30 | 1.06 | 0.96 | 0.98 |
| Pre | Perception of Data Literacy | 4.14 | 1.09 | 0.96 | 0.98 |
| Post | | 3.75 | 1.21 | 0.98 | 0.99 |
| Pre | Data Rationalization | 3.87 | 1.16 | 0.96 | 0.98 |
| Post | | 3.71 | 1.11 | 0.98 | 0.98 |

Notes: S.D.: standard deviation, $\alpha$: Cronbach's alpha, $\omega$: omega coefficient.

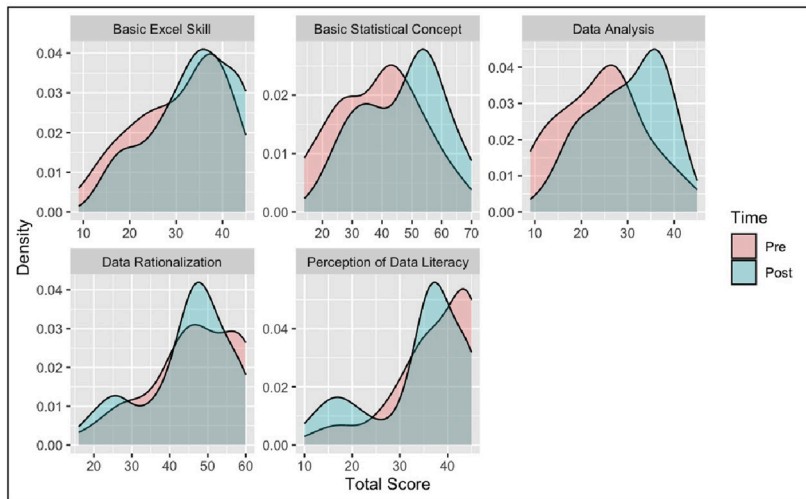

**Fig 4. Density distribution of the total score for the pre and post-survey based on students' perceptions of different categories.**

skills have significantly improved. In addition, perceptions about their basic Excel skills have improved only slightly. The only area in which there is no significant change is their perception of data literacy and their opinion of their ability to rationalise data.

We then address the main research question, "Did students' perceptions improve after the workshop, and if so, how large is the effect?" The concept category with the most significant effect size is data analysis. Because the scores are all based on ordered Likert scales, the Wilcoxon sign rank test was used. Based on Table 3 below, the signed test results revealed a significant improvement in students' perceptions of their understanding of basic statistical concepts and data analysis from pre- to post-test scores (the p-values are significant at the 5% level).

Following the findings presented above, we also conducted an analysis of gender differences in perceptions of data literacy. This analysis aims to identify potential factors contributing to variations in data literacy between genders [22]. Gender differences may encompass aspects such as educational background, professional experience, or even inherent cognitive abilities, which could potentially explain why one gender might exhibit a stronger grasp of data literacy. Previous studies have shown that there is a gender difference in STEM literacy [23]. By studying gender differences, it might provide evidence whether gender differences exist in data

**Table 3. Pre- and post- score of students' perceptions towards their understanding of statistical basic concepts.**

| Components | Pre Mean (S.D.) | Post Mean (S.D.) | Effect Size | p-value |
|---|---|---|---|---|
| Basic Statistical Concept | 2.77 (1.23) | 3.33 (1.15) | 0.281 | 0.000* |
| Basic Excel Skill | 3.43 (1.34) | 3.76 (1.11) | 0.131 | 0.269 |
| Data Analysis | 2.70 (1.28) | 3.30 (1.06) | 0.293 | 0.010* |
| Perception of Data Literacy | 4.14 (1.09) | 3.75 (1.21) | 0.160 | 0.343 |
| Data Rationalization | 3.87 (1.16) | 3.71 (1.11) | 0.084 | 1.000 |

Notes: S.D.: standard deviation; ** statistically significant at 5% level.

literacy. Researchers can better understand the factors contributing to differences in data analysis skills by studying gender differences in data literacy. They can also identify potential areas for improvement or intervention. Furthermore, understanding possible gender differences in data literacy can inform the development of educational and training programmes design to help individuals improve their data analysis skills, regardless of gender. Interestingly, Table 4 shows no statistically significant difference in perceptions on data literacy between male and female students. Both genders similarly perceive the importance of different aspects of data literacy.

It is generally believed that individuals who study science and mathematics have a more robust understanding of data literacy compared to those in other disciplines. This is because the concepts and methods of statistical analysis are fundamental to the curriculum in many science and mathematics fields. Students in these disciplines are often exposed to statistical concepts and techniques early on in their studies and are required to use them regularly in their coursework and research. As a result, they may be more familiar and proficient with data analysis techniques compared to students in other fields. Additionally, the nature of scientific and mathematical inquiry often requires the analysis and interpretation of data. Scientists and mathematicians are trained to be critical thinkers and approach problems logically and analytically. They are also taught to use statistical methods to draw conclusions and make predictions based on data. This emphasis on data analysis and interpretation may contribute to a more robust understanding of data literacy among science and maths students. Because of this, an important follow-up question arises: "Are there discipline-related differences in perceptions on data literacy"

Table 5 shows differences in perceptions based on subject disciplines. STEM majors exhibit a noticeable enhancement in their perception of their abilities in understanding basic statistical concepts, basic Excel skills and data analysis, as compared to their non-STEM counterparts (see Table 5).

## Qualitative findings

In both surveys, pre- and post-boot camps, students were asked about their understanding of data literacy and whether it changed after the boot camps in the post-survey. In the pre-survey, they were also asked about what they want to learn in the boot camp, while in the post-survey they were asked about what they have enjoyed in the boot camp. Following this, we conducted a comprehensive investigation of pre- and post-boot camp experiences through survey using both thematic analysis and sentiment analysis. Through thematic analysis, we identified recurring

**Table 4. Gender differences in perceptions towards understanding of statistical basic concepts.**

| Survey | Variable | Female Mean (S.D.) | Male Mean (S.D.) | Effect Size | *p*-value |
|--------|----------|--------------------|------------------|-------------|-----------|
| Pre | Basic Statistical Concept | 2.739(1.316) | 2.810(1.106) | 0.041 | 0.809 |
| Post | | 3.183(1.210) | 3.520(1.035) | 0.186 | 0.251 |
| Pre | Basic Excel Skill | 3.295 (1.335) | 3.605(1.325) | 0.152 | 0.349 |
| Post | | 3.609(1.139) | 3.951(1.044) | 0.205 | 0.205 |
| Pre | Data Analysis | 2.681(1.320) | 2.734(1.220) | 0.048 | 0.777 |
| Post | | 3.155(1.121) | 3.475(0.947) | 0.162 | 0.320 |
| Pre | Perception of Data Literacy | 4.092(1.193) | 4.191(0.956) | 0.075 | 0.784 |
| Post | | 3.928(1.190) | 3.531(1.191) | 0.075 | 0.332 |
| Pre | Data Rationalization | 3.873(1.248) | 3.856(1.044) | 0.046 | 0.648 |
| Post | | 3.699(1.122) | 3.713(1.104) | 0.158 | 0.649 |

**Table 5. Differences in perceptions towards understanding of statistical basic concepts based on disciplines.**

| Survey | Components | Science and Math Mean (S.D.) | Other Disciplines Mean (S.D.) | Effect Size | *p*-value |
|---|---|---|---|---|---|
| Pre | Basic Statistical Concept | 2.964(1.119) | 2.585(1.298) | 0.167 | 0.304 |
| Post | | 3.626(1.040) | 3.021(1.176) | 0.286 | 0.076** |
| Pre | Basic Excel Skill | 3.650(1.275) | 3.222(1.366) | 0.196 | 0.226 |
| Post | | 4.021(0.978) | 3.483(1.174) | 0.325 | 0.043* |
| Pre | Data Analysis | 2.806(1.273) | 2.608(1.274) | 0.032 | 0.855 |
| Post | | 3.603(0.829) | 2.972(1.174) | 0.309 | 0.051** |
| Pre | Perception of Data Literacy | 4.161(0.970) | 4.111(1.204) | 0.089 | 0.587 |
| Post | | 3.958(1.091) | 3.539(1.283) | 0.238 | 0.141 |
| Pre | Data Rationalization | 3.896(1.011) | 3.837(1.291) | 0.106 | 0.515 |
| Post | | 3.857(0.955) | 3.546(1.240) | 0.103 | 0.526 |

\* statistically significant at 5% level;

\*\* statistically significant at 10% level

themes and patterns in participant feedback, while sentiment analysis provided a quantifiable measure of the participants' emotional tone. The results show consistency across both methods validates our findings, highlighting significant trends and insights into participant experiences.

### Thematic analysis

Using a thematic analysis, we looked at students' responses to open-ended questions in the surveys to identify, analyse, and interpret patterns of meaning or themes [24] from the responses. We used an inductive approach in generating the themes, where we are open with the flow and influence of the data [25]. It can be said that the process of generating themes is through "data-driven" analysis [24].

From the analysis of both surveys, students' understanding of data literacy are mainly developed around the notion of data literacy as the ability to work with data or information. Some emerging themes from students' responses are including the ability to read, analyse and understand the data, the ability to think based on data, the ability to learn and understand data, the ability to communicate data and the ability to analyse and generate conclusions from the data. The other strands of themes include a range of abilities related to data literacy such as the ability to collect or gather information as data, the ability to manage data, the ability to process meaningful information, and the ability to visualise data. Responding to the sub question in the post-survey about the change of their perception of data literacy after joining the boot camps, students confirmed that they have better understanding about data literacy after joining the boot camp. They particularly highlighted increasing awareness about the importance of data literacy and practical insights from the intervention on how to work with data and to apply their conceptual understanding of data literacy.

In the pre-survey, students were asked what they expected to learn in the boot camp. Students showed their interest to learn data-related skills, such as how to analyse, present and manage the data, and practical skills of Microsoft Excel, such as using different formulas and functions. Nevertheless, few students still wanted to learn some basic statistical concepts, such as mean, median, mode and standard deviation. After the boot camp, in the post-survey, most students confirmed that the application of data-related skills and practical skills in using Microsoft Excel were parts of the boot camp they enjoyed learning the most. The practical insights such as how to collect, calculate, arrange, tidy up and process the data, as well as the

use of pivot table, filter, functions and formulas in Microsoft Excel were materials in the boot camp that students found useful. Two students particularly mentioned that the data challenge was the most insightful part of the boot camps. These students emphasised that the project in the data challenge provided the opportunity to implement their knowledge of data literacy from using the concepts for generating meaningful information and making decisions. With a real case from the activities in the boot camps, they gained an important insight of how data literacy is useful for daily life.

Details of themes and sub-themes generated from thematic analysis of the data from pre-boot camp survey are presented in Table 6 below.

Details of themes and sub-themes generated from thematic analysis of the data from post-boot camp survey are presented in Table 7 below.

Students who agreed to join a focused group discussion had the opportunity to elaborate their opinion about the boot camp and their perspective about data literacy in general. When they were asked about the general impression of the boot camp, all students agreed that it was useful as it provided them with applied skills related to some basic statistical concepts. A high-light is particularly attributed to practical insights they gained from the intervention, which they found complementary to what they have learned at school. A student said, "We have learned basic statistics, such as mean and median, but from the boot camps, we just knew the application to make decisions based on the data." Another student added, "For me, as at school, Excel was just introduced, so we just learned the basic skills, and joining the boot camps was very useful, so we knew the application. We just learned a few formulas at school, which were not discussed deeply".

## Sentiment analysis

To better contextualise this study with broader landscape in this data literacy educational program evaluation, sentiment analysis is able to offer a unique lens through which we can assess the benefit of such programs through the participant experience using their survey feedback.

**Table 6. Themes and sub-themes generated from thematic analysis of the data from pre-boot camp surveys.**

| Survey questions | Themes of responses |
|---|---|
| Describe your understanding the meaning of data literacy | • the ability to work with data or information<br>• the ability to read, analyse and understand data<br>• the ability to think based on data<br>• the ability to collect or gather information as data<br>• the ability to manage data<br>• the ability to process meaningful information<br>• the ability to communicate data |
| What would you like to learn in the boot camps? | • How to work with data<br>  ◦ how to analyse the data<br>  ◦ how to present the data<br>  ◦ working with data more efficiently<br>  ◦ creativity in working with data<br>  ◦ classify the data<br>  ◦ data management<br>• Using Microsoft Excel<br>  ◦ Functions in excel<br>  ◦ Formulas in excel<br>  ◦ Learn spreadsheet<br>• Basic of statistics<br>  ◦ Mean<br>  ◦ Median<br>  ◦ Mode<br>  ◦ Standard deviation |

**Table 7. Themes and sub-themes generated from thematic analysis of the data from post-boot camp surveys.**

| Survey questions | Themes of responses |
|---|---|
| Describe your understanding the meaning of data literacy | • an ability to work with data to generate meaningful information<br>• the ability to calculate data<br>• a set of individual ability and skill in reading, writing, communicating, counting and problem solving<br>• it is about managing data<br>• the ability to read, understand, create and communicate data<br>• the ability to understand the data<br>• the ability to visualize data<br>• the ability to read and analyse data<br>• It is about how we look at data<br>• an ability to obtain information from written data<br>• the ability to read, understand, create, and communicate data<br>• it is about data manipulation and extract information from it |
| Has it changed after joining the boot camps | • Yes<br>• I know and understand better<br>• Yes, I understand more about data literacy<br>• Data literacy is very important<br>• Data literacy is really interesting and useful<br>• Data literacy is critical for entering workforce<br>• I become more aware of a lot of information<br>• I understand more about steps to process the data<br>• No, but I can work on data faster and more effectively<br>• No, but I can do data calculation very quickly |
| What have you enjoyed learning the most at the boot camps? | • Working with data<br> • how to arrange data<br> • how to process data<br> • tidy up data<br> • knowledge of data, collecting data, calculating data<br> • creating data<br> • comparing data<br> • techniques to speed up data processing<br> • when I do the filter for data<br>• Skills in using Microsoft Excel<br> • Pivot table<br> • Different functions in excel<br> • Using formulas in excel<br> • Filtering data in excel<br> • How excel works<br>• Data challenge<br> • work on data challenge<br> • the project<br> • the implementation of data literacy<br> • problem solving<br>• Basic concepts of statistics<br> • types of data<br> • knowledge about data |

Through sentiment analysis, we are able to capture the nuances of participant experiences that traditional quantitative measures might overlook. Sentiment analysis is used in this study to explore the emotional and sentiment dynamics within textual data from the pre and post-boot camp survey responses. The sentiment analysis is conducted by utilising the NRC sentiment algorithm. Basically each response from students will be independently evaluated and a rating will be given for each response. Each word in the text is compared against a lexicon where words are associated with the sentiment scores. For example, in the NRC lexicon, words are tagged with eight different emotions (e.g., joy, anger, fear, anticipation, trust, surprise, sadness, and disgust) and binary sentiments (positive or negative). The sentiment scores of all the words in the sentence are summed to produce an overall sentiment score for the sentence.

In this study, we enhance the comprehensiveness of the NRC emotion lexicon by incorporating additional positive and negative words. This is to address gaps in sentiment analysis and improve the lexicon's overall accuracy in capturing emotional nuances. A list of added positive words and negative words are given in Table 8 below, in addition to the current NRC lexicon.

Fig 5 shows the analysis of the density distribution of the pre- and post-boot camp, it reveals a significant improvement in participants' attitudes. Before the boot camp, although all the scores are positive, the sentiment scores are more concentrated on the lower scores region. In contrast, post-boot camp sentiment scores show a notable increase. This shift in sentiment scores highlights the boot camp's positive impact on participants' perceptions and experiences.

## Discussion

The main objective of this paper is to investigate high school students' perceptions of their understanding of basic statistical concepts and data literacy skills after joining data literacy boot camps as an intervention. It also analyses the impact of the intervention and the perceived benefits of the skills learnt from the boot camp. The analysis of quantitative data shows that the student's perception of their understanding of basic statistical concepts and data analysis skills have significantly improved after the intervention, while basic Excel skills have improved slightly. As shown in Fig 4 above, most students perceived their understanding of basic statistical concepts was improved after joining the boot camp. As confirmed by participants of the focus group discussion, using real data from a real-life problem and context helps them to grasp the concepts more deeply. These findings confirm the recommendations of the GAISE College Report [14] that suggest the integration of real data in a particular context for a specific purpose in teaching introductory statistics. This integration, in turn, can help learners "develop the ability to think statistically" [14] (p. 8).

In general, participants confirmed that their experience joining the boot camp helped them to raise their awareness about the importance of data literacy, which indicated that the objective of the intervention was achieved. Responses in the post-survey confirmed that their perception of data literacy has changed after the boot camp, emphasising increasing awareness of

**Table 8. Additional lists of NRC lexicon.**

| Sentiment | Words |
|---|---|
| Positive | • learning<br>• help<br>• ability<br>• understand<br>• learn<br>• skill<br>• analyze<br>• answer<br>• able<br>• inform<br>• easier<br>• managing<br>• know<br>• yes<br>• can<br>• better<br>• good<br>• easily<br>• quickly<br>• important |
| Negative | • complicated<br>• confusing |

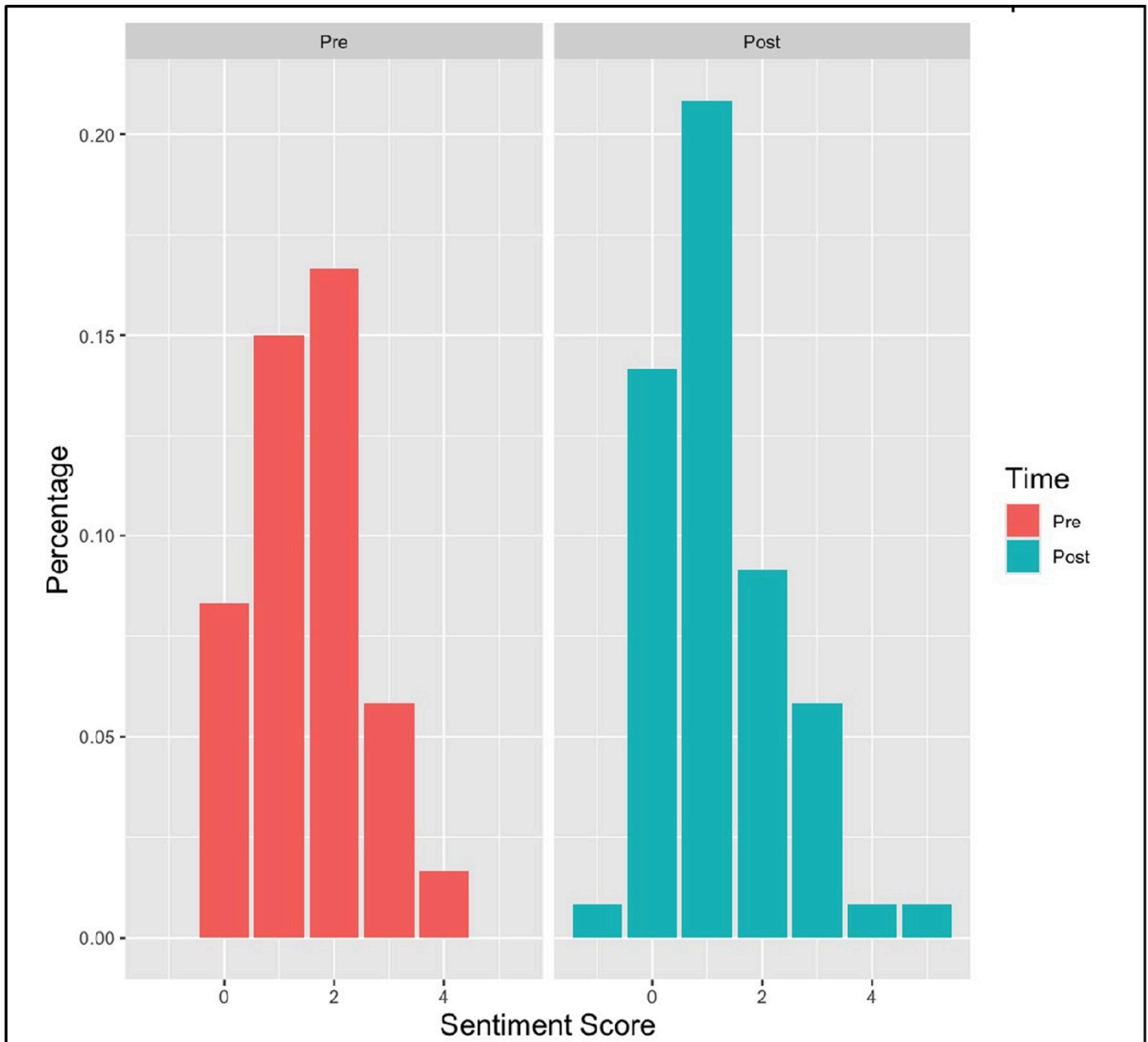

**Fig 5. Density distribution of the final sentiment scores pre-post bootcamp.**

the importance of data literacy in daily life. They became more aware of the availability and abundance of data in their surroundings [10]. In the FGD one student confirmed, "The boot camp was very good; for me, I can understand the use of data literacy for everyday life". This awareness is something that generally is lacking among science students [10]. Also, the shift in sentiment scores towards more positive for the pre and post-boot camp underlining boot camp's effectiveness in enhancing overall participant experience.

When students were asked in the FGD about their experience of joining the boot camp, one of the students said, "We have learned basic statistics, such as mean and median at school, but from the boot camps, we knew how to apply it when making decisions based on the data". Another student added, "For me, at school, Excel was just introduced, so we just learned the basic skills, and joining the boot camps was very useful, we knew how to apply it. We just learned a few formulas at school, which were not discussed deeply". These findings show that the design of the boot camps with materials and delivery focused on understanding basic concepts and their application is relevant for high school students, as suggested in the GAISE II [13]. The students' newfound confidence in their statistical skills is a testament to the effectiveness of the intervention.

Using real data sets in the boot camp also helps students to understand the relevance of the concepts in their daily lives. This is relevant to recommendations from the GAISE College Report [14] that using real data sets help students have better conceptual understanding. This strategy was particularly prominent in the data challenge provided to participants at the end of the boot camp after all sessions had been delivered. Students reflected on the experience working on the data challenge as challenging yet exciting and insightful. They confirmed that the experience allowed them to practise decision-making informed by data.

Improvement in using different functions and formulas of Microsoft Excel is another practical skill that students find insightful and useful from the boot camps. Although the quantitative analysis showed only a slight improvement in students' perceived ability to use Microsoft Excel, the qualitative data indicated that students found the practice of using the tool very useful and this was the part they enjoyed the most in the boot camp. One of the FGD participants stated, "I just knew how powerful it (pivot table) was, it can process vast amounts of data very quickly". As outlined in the GAISE report [14], using technology to analyse data is a main recommendation for teaching introductory statistics courses in higher education.

## Conclusions

Data literacy boot camps as an intervention to raise awareness of the criticality of data literacy skills are beneficial for students in secondary education. The findings of this study show that students perceive the intervention as complementing the understanding of some basic statistical concepts they have learned at school, as the boot camps improve their skills in applying the concepts to a real case and data set. Students had the opportunity to exercise their skills to assess the validity of data and how to use data to draw meaningful conclusions. Their ability to derive meaningful conclusions from data is evidence of the effectiveness of the intervention.

Based on the positive feedback and perceived benefits reported by students, it is recommended that data literacy boot camps be considered a strategic intervention to help secondary students transition smoothly to higher education and future careers. Additionally, teachers in high schools can incorporate these learning materials and approaches when teaching statistical concepts. From a policy perspective, this study suggests stronger advocacy to encourage schools to integrate data literacy into their curriculum.

Since this was a pilot project, there is room for improvement in the intervention's design and the scalability. For the design, this pilot project can be contextualised in terms of its content and delivery method to match students' needs and conditions. For scalability, there is an excellent opportunity to broaden the scope and coverage of the project to reach more secondary students in different contexts, in different regions in Indonesia and even adopted for contexts in different countries.

## Supporting information

**S1 Data.**

(XLS)

## Acknowledgments

We would like to acknowledge all students, teachers, and schools who have participated in this research project for their contribution to the project.

## Author Contributions

**Conceptualization:** Charanjit Kaur, Nurjannah Nurjannah, Ririn Yuniasih.

**Data curation:** Pei P. Tan.

**Project administration:** Ririn Yuniasih.

**Visualization:** Pei P. Tan.

**Writing – original draft:** Charanjit Kaur, Pei P. Tan, Ririn Yuniasih.

**Writing – review & editing:** Nurjannah Nurjannah, Ririn Yuniasih.

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
