## [Decision Letter · Decision Letter 0]

2 Jul 2024

PONE-D-24-17279Exploring data literacy self-perception among Indonesian high school studentsPLOS ONE

Dear Dr. Yuniasih,

Thank you for submitting your manuscript to PLOS ONE. After careful consideration, we feel that it has merit but does not fully meet PLOS ONE’s publication criteria as it currently stands. Therefore, we invite you to submit a revised version of the manuscript that addresses the points raised during the review process.

We look forward to receiving your revised manuscript.

Kind regards,

Muhammad Arsyad Subu, Ph.D

Academic Editor

PLOS ONE

Reviewers' comments:

Reviewer's Responses to Questions

**Comments to the Author**

1. Is the manuscript technically sound, and do the data support the conclusions?

Reviewer #1: Partly

Reviewer #2: Yes

2. Has the statistical analysis been performed appropriately and rigorously? 

Reviewer #1: No

Reviewer #2: Yes

3. Have the authors made all data underlying the findings in their manuscript fully available?

Reviewer #1: No

Reviewer #2: Yes

4. Is the manuscript presented in an intelligible fashion and written in standard English?

Reviewer #1: Yes

Reviewer #2: Yes

5. Review Comments to the Author

Reviewer #1: Please see the comments in attached file, authors need to clearly stated that design of this study, instruments used and whether authors developed the instruments (any theory used to develop) or adopted from similar studies, also qualitative analysis need further details especially in presenting the qualitative results. It should match with data analysis. Argumentation in discussion section need to be improved by citation to similar or relevant studies.

Reviewer #2: This is important study. The author might want to describe more clearly how the bootcamp was delivered. It stated in the manuscript that it was 2 virtual sessions, @ 2 hours. However it isn't clear yet who were the speakers, how specific topics were distributed and delivered in the first and second session? was it delivered during a school year period or in a school break? considering the school timeline/schedule which usually have determined by the school.

How the authors justify putting male and female students in one group of FGD?

6. PLOS authors have the option to publish the peer review history of their article (what does this mean?). If published, this will include your full peer review and any attached files.

Reviewer #1: No

Reviewer #2: No

---

## [Author Response · Author response to Decision Letter 0]

29 Aug 2024

Dear editor and the esteemed reviewers,

Thank you for giving me the feedback to our manuscript. We have modified my essay based on suggestions from reviewers.

Reviewer 1

Thank you for your feedback and encouraging words about the strength of the paper.

We have revisited the introduction and made some changes as follow:

• A methodology section has been added in the manuscript, explaining the study in general including the research design, methods of data collection and data analysis (Page 6 line 150).

• Also, explanations related to recruitment of participants and data literacy boot camps provided as sub sections ‘participants’ (Page 7 line 175).

• Further explanation of how instrument used in this study was developed and the concept and theoretical framework being adopted has been provided in the sub section ‘data collection’ under methodology section (Page 7 line 262)

• Explanation about Focus Group Discussion is provided in the methodology section (Page 7 line 167)

• More details of qualitative analysis have been added (Page 17 line 415)

• Further discussions with added literature have been added to the manuscript as recommended

Reviewer 2

Thank you for your comments and suggestions. We have applied your recommendations to improve the clarity of the article. Further information about Focus Group Discussion is provided in the methodology section (Page 7 line 167). A subsection of data literacy boot camps has been added under the methodology section, explaining details of the boot camps and addressing the reviewer’s comments. (Page 10 line 220)

---

## [Editor Report · Decision Letter 1]

9 Sep 2024

Exploring data literacy self-perception among Indonesian high school students

PONE-D-24-17279R1

Dear Dr. Yuniasih,

We’re pleased to inform you that your manuscript has been judged scientifically suitable for publication and will be formally accepted for publication once it meets all outstanding technical requirements.

Kind regards,

Muhammad Arsyad Subu, Ph.D

Academic Editor

PLOS ONE

Additional Editor Comments (optional):

Dear authors,

Thank you so much for responding the reviewer and editor comments.

Good luck
---

## [Editor Report · Acceptance letter]

31 Oct 2024

PONE-D-24-17279R1 

PLOS ONE

Dear Dr. Yuniasih, 

I'm pleased to inform you that your manuscript has been deemed suitable for publication in PLOS ONE. Congratulations! Your manuscript is now being handed over to our production team.

Kind regards, 

on behalf of

Dr. Muhammad Arsyad Subu 

Academic Editor

PLOS ONE